# Influence of Antenna Element Position Deviation on Radiation Performance of Helical Antenna Array

**DOI:** 10.3390/s23104827

**Published:** 2023-05-17

**Authors:** Lingqi Zeng, Chengxin Wang, Haibo Liu, Ning Liu, Jinxin Wang, Lanlan Zhou, Yong Wang, Yongqing Wang

**Affiliations:** 1State Key Laboratory of High-Performance Precision Manufacturing, Dalian University of Technology, Dalian 116024, China; 12104031@mail.dlut.edu.cn (L.Z.); 2872050935@mail.dlut.edu.cn (C.W.); hbliu@dlut.edu.cn (H.L.); wjx27@mail.dlut.edu.cn (J.W.); yqwang@dlut.edu.cn (Y.W.); 2School of Electrical Engineering, Dalian University of Technology, Dalian 116024, China; 3China Academy of Space Technology, Xi’an 710199, China; zhoull@mail.cast504.com (L.Z.); wangy26@mail.cast504.com (Y.W.)

**Keywords:** helical antenna, position deviation, feed array, radiation electrical properties

## Abstract

The electrical performance of the feed array is degraded because of the position deviation of the array elements caused by manufacturing and processing, which cannot meet the high performance feeding requirements of large feed arrays. In this paper, a radiation field model of the helical antenna array considering the position deviation of array elements is proposed to investigate the influence law of position deviation on the electrical performance of the feed array. With the established model, the rectangular planar array and the circular array of the helical antenna with a radiating cup are discussed and the relationship between electrical performance index and position deviation is established by numerical analysis and curve fitting method. The research results show that the position deviation of the antenna array elements will lead to the rise of the sidelobe level, the deviation of the beam pointing, and the increase of the return loss. The valuable simulation results provided by this work can be used in antenna engineering, guiding antenna designers to set optimal parameters when fabricating antennae.

## 1. Introduction

The feed array is one of the main core components of the geostationary orbit satellite, which provides feed power for satellite mobile communication signal transmission and reception and is known as the heart of the satellite antenna. The electrical properties of the integrated feed array are extremely sensitive to structural deviations caused by the machining errors during the manufacturing process. The unreasonable errors will reduce the electrical performance of the antenna, such as gain loss, sidelobe level lift, and beam position shift [1,2,3]. This seriously restricts the realization of high performance of the antenna, such as high gain, high pointing accuracy, and low sidelobe level. The higher the frequency, the more severe the effects [4]. However, the influence mechanism of the structural deviation of the feed array on the electrical performance is complex, and the relationship is unknown. Therefore, it is important to study the relationship between antenna structure error and electrical performance to provide a basis for the design of the array antenna structure and the manufacture of its tolerances.

In the research on the influence of errors on the antenna radiation performance, most researchers mainly focus on the feed error [5,6,7,8,9,10], thermal error [11,12,13,14,15,16], and antenna front structure deformation [17]. Few studies have been conducted to investigate the effect of array element processing position deviation on the electrical performance of antennae.

For the analysis of the influence of the array element position deviation on the electrical performance of the antenna, the typical research methods mainly include the probability statistics method [18,19,20] and the interval analysis method [11,21,22]. Reference [18] used the method of probability and statistics to study the influence of the position deviation of the array element on the electrical performance of the antenna. However, this method requires a large number of repeated calculations to obtain the statistical performance of the antenna. Wang [19] studied the electrical performance of the planar phased array antenna with an error by using the method of probability and statistics. In [20], when analyzing the influence of radiation element deviation on the electrical performance of the antenna, it is assumed that the error obeys a normal distribution, but an accurate mapping between the machining position deviation and the electrical performance is not established. However, the probability density distribution cannot always be obtained by using the method of probability statistics in practical engineering. The interval analysis method gives the upper and lower bounds of the error according to the interval algorithm and calculates the upper and lower intervals of the electrical performance of the antenna. The worst case method [11] was proposed based on the Cauchy–Schwarz inequality; it can predict the worst boundary of the performance interval. To improve the interval accuracy, Anselmi et al. [21,22] predicted the influence of excitation amplitude deviation on radiation patterns based on interval analysis. However, limited by the problems of interval expansion and algorithm accuracy, it is not suitable for establishing tight bounds in terms of causing changes in electrical properties of the high frequency array antenna. In particular, taking the position deviation of the array element into consideration, the acquisition of tight bounds is more difficult.

In addition, in the existing research, the antennae used are all simple dipole or loop antennae, and there are almost no reports on the influence of the position deviation of the array element on the performance of the helical antenna with a radiating cup.

To analyze the influence of the deviations generated in the manufacture of the integrated feed array on the electrical performance of the antenna, this paper establishes the radiation field of the helical antenna with radiation cups based on the geometric diffraction theory. Taking the rectangular plane array and the circular array as examples, based on the pattern product theorem, the total field model of the helical array antenna is established by using the perturbation analysis method. Changes in the antenna electrical performance indices before and after introducing the deviation are analyzed, as is the trend of the influence of the structural deviation on the antenna electrical performance.

This paper is organized as follows: Section 2 establishes the radiation field model of the helical antenna array considering the position deviation; Section 3 presents the results and analysis; Section 4 is the conclusion.

## 2. Modeling of the Radiation Field

Firstly, the radiation field and pattern function of the helical antenna unit with a radiating cup is created using the geometric diffraction theory and the radiation field of the axial mode helical antenna presented by Klaus [23]. The entire radiation field of the helical antenna rectangular planar array and the circular array is then established using the coordinate position connection and the pattern product theorem. The radiation field error model of the helical antenna array taking a position deviation into consideration is constructed. The position deviation of the antenna element is equivalent to the corresponding phase error, which is included in the initial phase of the antenna, the radiation field error model of the helical antenna array considering position deviation is established.

### 2.1. Modeling of Radiation Field of Helical Antenna Array Elements with Radiation Cups

As discussed in [24], the strict expression of the radiation field of the actual helical antenna can be obtained by regarding the current between helical wires as the transmission of constant amplitude traveling wave and obtaining a strict expression of the radiation field of an actual helical antenna. The electric field at the far region *P* can be represented by $E_{\phi }$ and $E_{\theta }$ components.
(1)$$\begin{array}{ll} E_{\phi } & =E_{0}\left\{J_{0}(z)\left[\frac{\sin(\nu +1)\phi_{0}}{\nu +1}+\frac{\sin(\nu -1)\phi_{0}}{\nu -1}\right]\right.\\ & +\sum_{n=1}^{\infty }i^{n}J_{n}(z)\left[\frac{\sin(\nu +n+1)\phi_{0}}{\nu +n+1}+\frac{\sin(\nu +n-1)\phi_{0}}{\nu +n-1}\right.\\ & \left.\left.+\frac{\sin(\nu -n+1)\phi_{0}}{\nu -n+1}+\frac{\sin(\nu -n-1)\phi_{0}}{\nu -n-1}\right]\right\} \end{array}$$
(2)$$\begin{array}{ll} E_{\theta } & =E_{0}\left\{2\tan\alpha \sin\theta \left[J_{0}(z)\frac{\sin\nu \phi_{0}}{\nu }\right.\right.\\ & \left.+\sum_{n=1}^{\infty }i^{n}J_{n}(z)\left(\frac{\sin(\nu +n)\phi_{0}}{\nu +n}+\frac{\sin(\nu -n)\phi_{0}}{\nu -n}\right)\right]\\ & +\cos\theta \left[-iJ_{0}(z)\left(\frac{\sin(\nu +1)\phi_{0}}{\nu +1}-\frac{\sin(\nu -1)\phi_{0}}{\nu -1}\right)\right.\\ & +\sum_{n=1}^{\infty }i^{n-1}J_{n}(z)\left(\frac{\sin(\nu +n+1)\phi_{0}}{\nu +n+1}-\frac{\sin(\nu +n-1)\phi_{0}}{\nu +n-1}\right.\\ & \left.\left.\left.+\frac{\sin(\nu -n+1)\phi_{0}}{\nu -n+1}-\frac{\sin(\nu -n-1)\phi_{0}}{\nu -n-1}\right)\right]\right\} \end{array}$$

In practical applications, it is expected that the helical antennae radiate in one direction. Thus, a practical helical antenna has a back cavity. The geometric size of the radiation cup and its coordinate system is shown in Figure 1. Due to the existence of the radiation cup, there must be a diffraction field at the edge of the diameter of the radiation cup, as shown in Figure 2. Actually, the radiation field of the helical antenna is the superposition of the direct field, the reflected field, and the diffracted field. In this paper, starting with the radiation field of the unit helical antenna, the geometric diffraction theory (GTD) is applied to calculate the radiation field of the helical antenna.

(1) Direct Field

The direct field expression along the Z-axis of the helical antenna with a radiating cup is:(3)$${\vec{E}}^{i}=\left(E_{\theta }^{i}\hat{\theta }+E_{\phi }^{i}\hat{\phi }\right)\exp\left(-jkR_{0}\right)/R_{0}\;0\leq \theta \leq \alpha_{1}$$
where $E_{\phi }^{i}$ and $E_{\theta }^{i}$ are calculated by (1) and (2), respectively. $\alpha_{1}=arctg\left(R_{d}/h_{0}\right)$, k=2π/λ.

(2) Radiation Cup Bottom Reflection Field

Let the reflection coefficients for the θ and ϕ components be $R_{\theta }$ and $R_{\phi }$, and the propagation coefficients be $k_{\theta }$ and $k_{\phi }$, respectively. The reflected field is equivalent to starting from Z=−2h. The two components of the reflected field at the bottom of the radiation cup are
(4)$$\begin{array}{c} \left\{\begin{array}{l} \begin{array}{ll} E_{\theta }^{r}= & R_{\theta }E_{\theta }^{i}(\theta )\exp\left(-j\left(k_{\theta }-k\right)2h/\cos\theta \right)\\ & \exp(-j2kh\cos\theta )\exp\left(-jkR_{0}\right)/R_{0} \end{array}\\ \begin{array}{ll} E_{\phi }^{r}= & R_{\phi }E_{\phi }^{i}(\theta )\exp\left(-j\left(k_{\phi }-k\right)2h/\cos\theta \right)\\ & \exp(-j2kh\cos\theta )\exp\left(-jkR_{0}\right)/R_{0} \end{array} \end{array}\right.\\ 0\leq \theta \leq \alpha_{2} \end{array}$$
where $\alpha_{2}=arctg\left[R_{d}/\left(2h+h_{0}\right)\right]$; $E_{\theta }^{i}(\theta )$, $E_{\phi }^{i}(\theta )$ is the incident field components in the θ and ϕ directions.

(3) Diffraction Field Generated by the Direct Field at the Edge of $Q_{1}$.

According to the geometric diffraction theory, the components of the direct field in the $\hat{\theta }$ and $\hat{\phi }$ directions can be derived as
(5)$$\begin{array}{c} \left\{\begin{array}{l} \begin{array}{ll} E_{\theta 1}^{d}= & E_{\theta }^{i}\left(Q_{1}\right)D_{h}\left(Q_{1}\right)\left(\exp\left(-jkR_{1}\right)/R_{1}\right){\left(R_{d}/\sin\theta \right)}^{1/2}\\ & \exp\left(jkR_{d}\sin\theta \right)\exp\left(-jkR_{0}\right)/R_{0} \end{array}\\ \begin{array}{ll} E_{\phi 1}^{d}= & E_{\phi }^{i}\left(Q_{1}\right)D_{s}\left(Q_{1}\right)\left(\exp\left(-jkR_{1}\right)/R_{1}\right){\left(R_{d}/\sin\theta \right)}^{1/2}\\ & \exp\left(jkR_{d}\sin\theta \right)\exp\left(-jkR_{0}\right)/R_{0} \end{array} \end{array}\right.\\ 0^{\circ }\leq \theta \leq 180^{\circ }\;\mathrm{and}\;270^{\circ }\leq \theta \leq 360^{\circ } \end{array}$$
where $E_{\theta }^{i}\left(Q_{1}\right)$ and $E_{\phi }^{i}\left(Q_{1}\right)$ are the θ and ϕ components of the direct field at the diffraction point $Q_{1}$, respectively; $D_{h}\left(Q_{1}\right)$ and $D_{s}\left(Q_{1}\right)$ is the hard boundary and soft boundary diffraction coefficients at the diffraction point $Q_{1}$, respectively; $R_{1}=\sqrt{R_{d}^{2}+h_{0}^{2}}$. Similarly, the two components $E_{\theta 2}^{d}$ and $E_{\phi 2}^{d}$ of the diffraction field generated by the direct field $Q_{2}$ can be obtained.

(4) Reflected Field Generated by a Direct Field on the Left Sidewall

According to the reflection theory, the two components of the reflected field produced by the direct field at the left side wall are as follows:(6)$$\begin{array}{c} \left\{\begin{array}{l} \begin{array}{ll} E_{\theta 2}^{r}= & -jE_{\theta }^{i}(\theta )\exp\left(-jk\left(2R_{d}\sin\theta +h_{0}\cos\theta \right)\right)\\ & \exp\left(-jkR_{0}\right)/R_{0} \end{array}\\ \begin{array}{ll} E_{\phi 2}^{r}= & jE_{\phi }^{i}(\theta )\exp\left(-jk\left(2R_{d}\sin\theta +h_{0}\cos\theta \right)\right)\\ & \exp\left(-jkR_{0}\right)/R_{0} \end{array} \end{array}\right.\\ \alpha_{1}\leq \theta \leq \alpha_{3} \end{array}$$
(7)$$\begin{array}{c} \left\{\begin{array}{l} \begin{array}{ll} E_{\theta 2}^{r}= & -jE_{\theta }^{i}(\theta )\exp\left(-jk\left(2R_{d}\sin\theta +h_{0}\cos\theta \right)\right)\\ & \exp\left(-jkR_{0}\right)/R_{0} \end{array}\\ \begin{array}{ll} E_{\phi 2}^{r}= & jE_{\phi }^{i}(\theta )\exp\left(-jk\left(2R_{d}\sin\theta +h_{0}\cos\theta \right)\right)\\ & \exp\left(-jkR_{0}\right)/R_{0} \end{array} \end{array}\right.\\ \alpha_{1}\leq \theta \leq \alpha_{3} \end{array}$$

In the equation, $\alpha_{3}=arctg\left(3\frac{R_{d}}{h_{0}}\right)$.

(5) Diffraction Field Generated a by the Reflection Field from the Left Wall

According to GTD, the two components of the diffraction field generated by the left wall reflection field $Q_{1}$ can be derived as follows:(8)$$\begin{array}{c} \left\{\begin{array}{l} \begin{array}{ll} E_{\theta r1}^{d}= & E_{\theta 2}^{r}\left(Q_{1}\right)D_{h}\left(Q_{1}\right){\left(R_{d}/\sin\theta \right)}^{1/2}\\ & \exp\left(jk\left(R_{d}\sin\theta +h_{0}\cos\theta \right)\right)\exp\left(-jkR_{0}\right)/R_{0} \end{array}\\ \begin{array}{ll} E_{\phi r1}^{d}= & E_{\phi 2}^{r}\left(Q_{1}\right)D_{s}\left(Q_{1}\right){\left(R_{d}/\sin\theta \right)}^{1/2}\\ & \exp\left(jk\left(R_{d}\sin\theta +h_{0}\cos\theta \right)\right)\exp\left(-jkR_{0}\right)/R_{0} \end{array} \end{array}\right.\\ \;\;0^{\circ }\leq \theta \leq 180^{\circ }\;\mathrm{and}\;270^{\circ }\leq \theta \leq 360^{\circ } \end{array}$$

$E_{\theta 2}^{r}$ and $E_{\phi 2}^{r}$ are calculated by (8), as follows:(9)$$\left\{\begin{array}{l} E_{\theta 2}^{r}\left(Q_{1}\right)=-jE_{\theta }^{r}(\theta )\left(\sqrt{3}/R_{2}\right)\exp\left(-jkR_{2}\right)\\ E_{\phi 2}^{r}\left(Q_{1}\right)=jE_{\phi }^{r}(\theta )\left(\sqrt{3}/R_{2}\right)\exp\left(-jkR_{2}\right) \end{array}\right.$$

In the equation, $R_{2}=\sqrt{9R_{d}^{2}+h_{0}^{2}}$.

Similarly, the calculation equations of the right wall reflection fields $E_{\theta 1}^{r}$ and $E_{\phi 1}^{r}$ the diffraction fields $E_{\theta r2}^{d}$ and $E_{\phi r2}^{d}$ generated by the right wall reflection field $Q_{2}$ can be derived. The high order diffraction field is small and can be ignored.

(6) Element Radiation Field and Pattern Function

The total radiation field is a superposition of various fields:(10)$$\begin{array}{l} E_{\theta }=E_{\theta }^{i}+E_{\theta }^{r}+E_{\theta 1}^{d}+E_{\theta 2}^{d}+E_{\theta r1}^{d}+E_{\theta r2}^{d}+E_{\theta 1}^{r}+E_{\theta 2}^{r}\\ E_{\phi }=E_{\phi }^{i}+E_{\phi }^{r}+E_{\phi 1}^{d}+E_{\phi 2}^{d}+E_{\phi r1}^{d}+E_{\phi r2}^{d}+E_{\phi 1}^{r}+E_{\phi 2}^{r} \end{array}$$

The total radiation field is recorded as:(11)$${\vec{E}}_{e}\left(\theta ,\;\phi \right)=E_{\theta }\hat{\theta }+E_{\phi }\hat{\phi }$$

Further, the unit pattern function can be obtained from the unit radiation field, as shown in (10):(12)$$f\left(\theta ,\varphi \right)=\frac{\left|E_{e}\left(\theta ,\varphi \right)\right|}{E_{M}}$$

In the equation, $E_{M}$ is the maximum value of the field strength amplitude at a certain point in any direction.

### 2.2. Modeling the Radiation Field of Helical Antenna Array

(1) Rectangular Planar Array Radiation Field

There is a rectangular planar array of M×N elements placed in the X-Y plane, and a coordinate system is established as shown in Figure 3. The column spacing is dx, and the row spacing is dy. According to the spatial geometric relationship shown in Figure 3, the calculation equation for the angle of the far field target relative to the coordinate axis and the direction cosines is given as follows:(13)$$\left\{\begin{array}{l} \cos\alpha_{x}=\sin\theta \cos\varphi \\ \cos\alpha_{y}=\sin\theta \sin\varphi \\ \cos\alpha_{z}=\cos\theta \end{array}\right.$$

The unit vector from the coordinate origin to the observation point is expressed as
(14)$$\hat{r}=\hat{x}\sin\theta \cos\varphi +\hat{y}\sin\theta \sin\varphi +\hat{z}\cos\theta$$

The coordinate positions of the unit are
(15)$$\left\{\begin{array}{c} x_{m}=md_{x}\\ y_{n}=nd_{y} \end{array}\right.\;\begin{array}{c} ,\\ , \end{array}\;\;\begin{array}{c} m=0,1,2,\cdot \cdot \cdot ,M\\ \;n=0,1,2,\cdot \cdot \cdot ,N \end{array}$$

The position vector is
(16)$${\vec{\rho }}_{{}_{mn}}=\hat{x}md_{x}+\hat{y}nd_{y}$$

The spatial phase differences between adjacent antenna elements along the x-axis and y-axis are
(17)$$\left\{\begin{array}{l} \Delta \varphi_{x}=kd_{x}\cos\alpha_{x}\\ \Delta \varphi_{y}=kd_{y}\cos\alpha_{y} \end{array}\right.$$
where k is the number of wavelengths, k=2π/λ.

Therefore, the spatial phase difference of the unit $\left(m,n\right)$ relative to the coordinate origin is
(18)$$\Delta \varphi_{mn}=m\Delta \varphi_{x}+n\Delta \varphi_{y}$$

Assuming that the excitation current of the unit $\left(m,n\right)$ is ${\dot{I}}_{mn}$, the radiation field in the far region can be expressed as
(19)$$E_{mn}\left(\theta ,\varphi \right)=f\left(\theta ,\varphi \right){\dot{I}}_{mn}e^{j\Delta \varphi }$$

Further, the radiation field of the entire planar array is
(20)$$\begin{array}{ll} E_{T}=\sum_{m}\sum_{n}E_{mn} & =f\left(\theta ,\varphi \right)\sum_{m=0}^{M-1}\sum_{n=0}^{N-1}{\dot{I}}_{mn}e^{j\Delta \varphi_{mn}}\\ & =f\left(\theta ,\varphi \right)\sum_{m=0}^{M-1}\sum_{n=0}^{N-1}{\dot{I}}_{mn}e^{j\left(m\Delta \varphi_{x}+n\Delta \varphi_{y}\right)}\\ & =f\left(\theta ,\varphi \right)\sum_{m=0}^{M-1}\sum_{n=0}^{N-1}{\dot{I}}_{mn}e^{j\left(kmd_{x}\cos\alpha_{x}+knd_{y}\cos\alpha_{y}\right)} \end{array}$$

The matrix factor in the equation is recorded as
(21)$$S\left(\theta ,\varphi \right)=\sum_{m=0}^{M-1}\sum_{n=0}^{N-1}{\dot{I}}_{mn}e^{j\left(kmd_{x}\cos\alpha_{x}+knd_{y}\cos\alpha_{y}\right)}$$

(2) Circular Array Radiation Field

There are N units distributed on the circular array, as shown in Figure 4. The angle of the nth unit is $\varphi_{n}$, its position coordinate is $\left(x_{n},y_{n}\right)$, and the far-field radiation field of this unit is
(22)$$E_{n}={\dot{I}}_{n}f\left(\theta ,\varphi \right)e^{-jk\left(R_{n}-r\right)}$$
where ${\dot{I}}_{n}$ is the current excitation of the unit, $f\left(\theta ,\varphi \right)$ is the pattern of the unit, and the distance of the nth unit to the observation point is $R_{n}$.

The coordinates of its position for the nth unit are
(23)$$x_{n}=a\cos\varphi_{n},\;y_{n}=a\sin\varphi_{n}$$

Its position vector is
(24)$$\vec{\rho_{n}}=\hat{x}x_{n}+\hat{y}y_{n}=a\left(\hat{x}\cos\varphi_{n}+\hat{y}\sin\varphi_{n}\right)$$

The unit vector in the r direction is
(25)$$\hat{r}=\hat{x}\sin\theta \cos\varphi +\hat{y}\sin\theta \sin\varphi +\hat{z}\cos\theta$$

Then, the wave path difference from the nth unit to an observation $R_{n}$ in the far area and the distance r from the coordinate origin to the same observation point is
(26)$$\begin{array}{ll} R_{n}-r & =-\hat{r}\cdot \rho_{n}=-\left(x_{n}\cos\varphi +y_{n}\sin\varphi \right)\sin\theta \\ & =-a\left(\cos\varphi_{n}\cos\varphi +\sin\varphi_{n}\sin\varphi \right)\sin\theta \\ & =-a\sin\theta \cos\left(\varphi -\varphi_{n}\right) \end{array}$$

The total radiation field of the circular array is
(27)$$\begin{array}{ll} E & =\sum_{n}E_{n}=f\left(\theta ,\varphi \right)\sum_{n=1}^{N}I_{n}e^{j\left[ka\sin\theta \cos\left(\varphi -\varphi_{n}\right)+\alpha_{n}\right]}\\ & =S(\theta ,\varphi )f\left(\theta ,\varphi \right) \end{array}$$

The matrix factor in the equation is recorded as
(28)$$S\left(\theta ,\varphi \right)=\sum_{n=1}^{N}I_{n}e^{j\left[ka\sin\theta \cos\left(\varphi -\varphi_{n}\right)+\alpha_{n}\right]}$$

The beam at the maximum pointing $\left(\theta_{0},\varphi_{0}\right)$ satisfies the relation: $ka\sin\theta_{0}\cos\left(\varphi_{0}-\varphi_{n}\right)+\alpha_{n}=0$, we draw the following conclusions:(29)$$\alpha_{n}=-ka\sin\theta_{0}\cos\left(\varphi_{0}-\varphi_{n}\right)$$

Substituting (27) into (26) yields
(30)$$S(\theta ,\varphi )=\sum_{n=1}^{N}I_{n}e^{jka\left[\sin\theta \cos\left(\varphi -\varphi_{n}\right)-\sin\theta_{0}\cos\left(\varphi_{0}-\varphi_{n}\right)\right]}$$

### 2.3. Modeling the Radiation Field of Helical Antenna Array Considering the Position Deviation

(1) Rectangular Planar Array Radiation Field

The signal’s distance traveled when it reaches the receiving unit will change as a result of the unit’s positional deviation, which might result in a change in the signal’s arrival time. This timing skew can affect the quality and accuracy of the received signal, especially in applications that require precise measurements of the signal. In addition, the beam pointing of the antenna is usually designed to point in a specific direction in order to receive signals from that direction. Misalignment of the antenna position may cause the beam to point away from the intended position, resulting in a reduced quality of the received signal or a complete loss of signal. This effect can be equivalent to the phase error corresponding to the unit position deviation, which is included in the initial phase of the array antenna.

Suppose the position offset of the unit $\left(m,n\right)$ is $\left(\Delta x,\Delta y,\Delta z\right)$, then the spatial phase difference of the unit relative to the unit $\left(0,0\right)$ is
(31)$$\begin{array}{ll} \Delta \varphi_{mn} & =k\left(md_{x}+\Delta x_{mn}-\Delta x_{00}\right)\cos\alpha_{x}\\ & +k\left(nd_{y}+\Delta y_{mn}-\Delta y_{00}\right)\cos\alpha_{y}\\ & +k\left(\Delta z_{mn}-\Delta z_{00}\right)\cos\alpha_{z} \end{array}$$

In this study, the phase error in the antenna array and the position deviation in the Z direction caused by the phase shifter are not considered for the time being. The rectangular grid rectangular plane array pattern function is
(32)$$\begin{array}{ll} E_{T}=\sum_{m}\sum_{n}E_{mn} & =f\left(\theta ,\varphi \right)\sum_{m=0}^{M-1}\sum_{n=0}^{N-1}{\dot{I}}_{mn}e^{j\Delta \varphi_{mn}}\\ & =f\left(\theta ,\varphi \right)\sum_{m=0}^{M-1}\sum_{n=0}^{N-1}{\dot{I}}_{mn}\cdot \\ & e^{jk\left[\left(md_{x}+\Delta x_{mn}\right)\cos\alpha_{x}+\left(nd_{y}+\Delta y_{mn}\right)\cos\alpha_{y}\right]} \end{array}$$

(2) Circular Array Radiation Field

Assuming that the position offset of the unit $\left(m,n\right)$ is $\left(\Delta x,\Delta y,\Delta z\right)$, the offset unit position $\left({x^{\prime }}_{n},{y^{\prime }}_{n}\right)$ is determined by the radius a and angle $\varphi_{n}$ of the circular array, and we obtain
(33)$$\begin{array}{l} \left({x^{\prime }}_{n},{y^{\prime }}_{n}\right)=\left(x_{n}+\Delta x,y_{n}+\Delta y\right)\cdot \\ =\left[\left(a+\Delta a\right)\cos\left(\varphi_{n}+\Delta \varphi \right),\left(a+\Delta a\right)\sin\left(\varphi_{n}+\Delta \varphi \right)\right] \end{array}$$

Substituting (31) into (28), we discern that the circular array factor with deviation is
(34)$$S(\theta ,\varphi )=\sum_{n=1}^{N}I_{n}e^{jk\left(a+\Delta a\right)\left[\sin\theta \cos\left(\varphi -\varphi_{n}-\Delta \varphi \right)-\sin\theta_{0}\cos\left(\varphi_{0}-\varphi_{n}-\Delta \varphi \right)\right]}$$

## 3. Results and Discussion

In this section, a single winding axial helical antenna with a back cavity is designed, and the optimal electrical performance parameters are obtained through the optimized design. Taking the array as an example, the influence of the structural deviation on the radiated electrical performance of the antenna is analyzed by using the research method of the perturbation method and the full wave simulation software HFSS. The specific calculation process is shown in Figure 5.

### 3.1. Design and Analysis of Helical Antenna Elements

A single-wound axial helical antenna is an antenna composed of a single metal wire wound into a cylindrical helical shape. The schematic diagram of the structure is shown in Figure 1. The main parameters of the helical antenna include the diameter D of the helix, the circumference *C* of the helix, the pitch *S* of the helix, the length *L* of each week, the diameter of the circular ground plane $R_{d}$, the number of turns *N*, and the diameter *d* of the helical wire. One end of the helical antenna is connected with the inner conductor of the coaxial line, and we connect the outer conductor of the coaxial line with the circular ground plane. With the aid of the cylindrical ground plane, the current on the outer surface of the outer conductor of the coaxial line is reduced. Then, the change of the input impedance in the working frequency band can be reduced and the backward radiation will be suppressed.

In addition, the cylindrical ground plane has a certain shielding effect on the electromagnetic influence of the external environment. It took the diameter of the general ground plane as $a=\left(0.75-1.5\right)\lambda_{0}$ [11,21]. When $0.75\lambda_{0}\leq C\leq 1.3\lambda_{0}$ the maximum radiation direction is along the axial direction of the helix. A better axial ratio and gain can be obtained by optimizing the height, the number of turns, pitch, and several turns of the helix. The main size parameters of the single wound axial helical antenna are shown in Table 1.

The designed axial-mode helical antenna is simulated and calculated by HFSS. The center frequency of the designed antenna is f=1.89 GHz. The antenna designed in this paper has an axial ratio bandwidth of less than 3 dB in the range of 1.2–2 GHz. In the frequency sweep between 1.2–2 GHz, there are six calculation frequency points, which are 1.2 GHz, 1.4 GHz, 1.6 GHz, 1.8 GHz, and 2 GHz. We set the circular ground plane and the outer surface of the coaxial line as the ideal conductor boundary. To ensure that the phase of the field reflected from the floor is in phase with the forward wave and has the maximum gain, the radiation boundary distance of the helical antenna should be no less than 0.25 working wavelength from the radiator. The feeding method adopts coaxial line feeding to reduce the complexity of feeding; the grid division adopts adaptive grid division to auto-generate an accurate and effective grid; the wave port excitation is used; and the input impedance is 50 Ω.

The antenna’s electromagnetic wave energy is unevenly dispersed in space, so the antenna’s directional diagram is used to illustrate how the antenna’s radiation parameter changes depending on the spatial orientation. The three-dimensional pattern of the helical antenna unit designed in this paper is shown in Figure 6, and Figure 7 shows the gain curve of the rectangular coordinate antenna at the center frequency point f=1.89 GHz. We can see that the maximum gain of the antenna is greater than 10 dB, the 3 dB bandwidth is 44°, and it has a good cross polarization.

As shown in Figure 8, we give the normalized gain pattern at 1.2 GHz, 1.4 GHz, 1.6 GHz, and 1.8 GHz of the helical antenna. The gain of the left hand circularly polarized antenna is higher than that of the right handed circular polarization, and the antenna is a left-handed circularly polarized wave. Since the energy radiated from the plane above the radiation cup is higher than that radiated from the back, emitting the plane above the direction diagram is stronger than that from the bottom, so the ratio between front and rear is improved to some extent.

To investigate the matching degree, the return loss and VSWR are calculated. As shown in Figure 9, the return loss of the cylindrical ground surface is lower than −8.6 dB in the working frequency range, and the working center frequency is lower than −40 dB, which indicates that the designed cylindrical ground surface axial mode helical antenna has a good matching state. It can be seen from Figure 10 that the standing wave ratio of the cylindrical ground plane is 0.94 dB at the center frequency and 3 dB bandwidth from 1.2–3 GHz, showing a high matching degree and wide axial ratio bandwidth. In the low frequency band, the VSWR of the cylindrical ground surface is smaller than that of the circular ground surface, and the return loss is lower.

Figure 11 shows the variation curve of antenna gain with frequency for two types of ground planes. The gain of the helical antenna with a cylindrical ground plane is higher than that of a circular ground plane. The cylindrical ground plane can significantly improve the gain of the antenna. The shape of the ground plane conductor improves the axial ratio and reduces the size of the sidelobes.

### 3.2. Model Verification

To illustrate the validity of the developed model, we verify it by simulating a rectangular planar array with HFSS. Considering the computing performance of the computer, the number of rectangular planar array elements is 4, the unit spacing is λ/2, and the excitation of the array antenna adopts the same amplitude and in phase excitation.

As shown in Figure 12, the model proposed in this paper shows good agreement with the HFSS results in both the main lobe and side lobe regions. In $\phi =0^{\circ }$ and $\phi =90^{\circ }$, the main lobe gain and beam width of both are the same. In $\phi =0^{\circ }$, the first sidelobe shows good consistency. In $\phi =90^{\circ }$, the absolute value of the first sidelobe has a maximum difference of 0.98 dB. The reason for the deviation is that the HFSS software considers the influence of mutual coupling and the precision of the grid. The above results show that the proposed model is effective for analyzing the effect of the position deviation of the array elements on the electrical performance of the antenna.

### 3.3. Rectangular Plane Array

Considering the computing capacity of the actual computer, the number of array elements used in this study will be 2×2. Under constant amplitude in phase excitation, to analyze, more comprehensively, the influence of position deviation on the electrical performance of the antenna, this study takes the typical rectangular array and circular array as examples for analysis. A schematic diagram of a rectangular array is shown in Figure 13. For this array, the position deviation along a single direction of the antenna is mainly discussed. In the numerical simulations, the position deviation δ is ranging from 0 mm to λ/8 mm. The calculation acquires the three-dimensional pattern of the rectangular planar array antenna, change diagram, and sidelobe level change curve, as shown in Figure 14, Figure 15 and Figure 16. As observed, the maximum amplitude of the first side lobe close to the main lobe is larger than that of the other sidelobe in the far region. Therefore, the sidelobe level of the array is determined by its first sidelobe level, and the maximum of the left first sidelobe level or the right first sidelobe level is taken.

As shown in Figure 15 and Figure 16, the two dotted lines in Figure 15a are Figure 15b and Figure 15c respectively, and the two dotted lines in Figure 16a are Figure 16b and Figure 16c respectively. The following conclusions can be drawn: (1) The main lobe gain does not change significantly under different position deviations, and the gain changes at *ϕ* = 0° and *ϕ* = 90° phases are consistent. No matter *ϕ* = 0° or *ϕ* = 90°, the maximum gain of the main lobe fluctuates up and down, and the change is within the range of 0.065 dB; when δ=1/40λ mm it is more sensitive to the antenna gain, the antenna gain decreases by 0.013 dB. (2) Antenna position deviation has a significant effect on the sidelobe level. For *ϕ* = 0° and *ϕ* = 90°, the sidelobe level increases with the increase of position deviation, and when δ=1/8λ mm, the sidelobe levels of the two phases increase by 1.25 dB and 1.1 dB, respectively. The antenna position deviation also has a certain influence on the sidelobe level in the far area, which increases by 0.93 dB when ϕ = 0° and θ is between −148° to 170°. When *ϕ* = 90° and θ is between 125°~155°, it increased the antenna sidelobe level by 1.61 dB. We cannot ignore the influence of structural deviation on electrical performance. (3) Because of existing structural deviation, whether in *ϕ* = 0° or *ϕ* = 90°, the main lobe beam pointing offset is small and can be ignored, the side lobe pointing has shifted, and the maximum offset is 2°.

Figure 17 is the gain loss and beam width variation curves of the antenna under different antenna position deviations. The analysis shows that the larger the antenna position deviation, the greater the antenna gain loss. We use the linear fitting method to obtain the changing relationship between the gain loss and the structural deviation. The summarized relationship is
(35)$$\Delta G=0.00156\left(\pm 2.6\times 10^{-3}\right)+0.00342\left(\pm 2.2\times 10^{-4}\right)\delta$$

We can conclude that the beam width decreases with the increase in the deviation in the antenna structure, and the two are approximately linearly related. The summarized relationship is
(36)$$\begin{array}{ll} \Delta B & =\left(2.73775\pm 0.72922\right)\times \exp\left(-\delta /\left(19.85669\pm 8.94814\right)\right)\\ & +\left(19.09389\pm 0.77591\right) \end{array}$$

In Figure 18, it is found that at the center frequency, the greater the antenna position deviation, the greater the return loss, showing that the greater par of the energy was lost because of reflection during the transmission of electromagnetic waves, which has a very detrimental effect on the antenna’s electrical performance for radiation.

### 3.4. Circular Array

Similar to the rectangular planar array antenna, Figure 19 shows the schematic diagram of the circular array antenna. For the circular array antenna, the origin of the coordinate system is the center of the circular array, and the radius vector δ is ranging from 0 mm to 1/8λ mm to analyze the influence of the antenna position’s deviation on the radiated electrical performance. The three-dimensional pattern, the maximum gain variation curve of the antenna, and the antenna sidelobe level variation curve of the circular array antenna are obtained by HFSS, as shown in Figure 20, Figure 21 and Figure 22.

The two dotted lines in the red box in Figure 21a are Figure 21b and Figure 21c, respectively, from Figure 21 and Figure 22 the following conclusions can be drawn: (1) Due to design reasons, the maximum gain of the antenna is shifted *θ* = 0°, but this does not affect the analysis of the influence law of the radiation electrical performance of the antenna with positional deviation. Whether in *ϕ* = 0° or *ϕ* = 90°, the gain of the circular array antenna with position deviation decreases as the error increases. When *ϕ* = 90°, due to the existence of the position deviation, not only is the antenna gain reduced but the beam pointing offset also occurs, and the maximum offset reaches 17°, which has an extremely adverse effect on the electrical performance of the antenna. (2) Similar to the rectangular planar array antenna, the position deviation of the array element has a significant impact on the antenna’s sidelobe level. When *ϕ* = 0° and δ=1/10λ mm, the electrical performance of the antenna is the most sensitive, the sidelobe level lift reaches 2.32 dB, and the maximum beam pointing deviation is 12°. When *ϕ* = 90°, due to the existence of the position deviation, and $100^{\circ }\leq \theta \leq 180^{\circ }$, the maximum sidelobe level rise is 4.84 dB, and the beam shift is 8°, which cannot meet the stringent electrical performance requirements.

Figure 23 shows the variation curves of gain loss and beam width under different position deviations of the circular array antenna. Antenna gain loss and beam width versus skew are not linearly related, unlike rectangular planar arrays. We use logistic regression analysis to obtain the relationship between gain loss and position deviation, such as the following formula:(37)$$\begin{array}{ll} \Delta G & =\left(0.28407\pm 0.01167\right)\\ & +\frac{\left(-8.22391\times 10^{-4}\pm 0.00654\right)-\left(0.28407\pm 0.01167\right)}{1+{\left(\delta /\left(6.74904\pm 0.34048\right)\right)}^{\left(2.30327\pm 0.25891\right)}} \end{array}$$

We use Boltzmann regression analysis to obtain the relationship between beam width and position deviation:(38)$$\begin{array}{ll} \Delta B & =\left(58.85929\pm 0.84042\right)\\ & +\frac{\left(50.84991\pm 0.20005\right)-\left(58.85929\pm 0.84042\right)}{1+\exp\left(\left(\delta -\left(16.99439\pm 0.38765\right)\right)/\left(0.99427\pm 0.28531\right)\right)} \end{array}$$

When there is position deviation, the antenna gain loss increases continuously. When the deviation varies within 1/16λ mm, the beam width of the antenna varies within 2°. When the deviation is greater than 1/16λ mm, the antenna’s performance is negatively impacted by the sharp increase in beam width, which also causes the antenna directivity to deteriorate, limits the working distance of the antenna, weakens its capacity to block out interference, and worsens its working distance.

We obtained the variation curves of the return loss of the circular array antenna under different position deviations, as shown in Figure 24. The circular array and rectangular array have similar laws. With increased deflection, the antenna return loss increases, and when the deflection reaches 1/8λ mm, the return loss increases by 14.72 dB.

The positional deviation of the helical antenna element causes the antenna gain to decrease, the side lobe level to increase, and the main lobe pointing to deviate. Given the above results, the adverse consequences caused by the position deviation can be compensated by further optimizing the design of the axial mode helical antenna; for example, the diameter of the helical antenna conductor is usually between 0.005λ and 0.05λ, and the gain of the antenna can be improved by optimizing the design. The helix radius will affect the resonant frequency of the helix antenna. By changing the radius of the helix, a better match with the design frequency can be achieved [25]. The best electrical performance can also be obtained by changing the geometry of the antenna, such as using a dual-arm hemispherical helical antenna [26].

## 4. Conclusions

In this paper, based on the geometric diffraction theory and linear fitting method, a radiation field model of the helical antenna array considering the position deviation of array elements is established. Taking the typical rectangular planar array and circular array as examples, the influence of the processing position deviation of the antenna array element on the electrical performance of the feed array is analyzed, and the mapping relationship between the array antenna gain loss, beam width, and machining position deviation is revealed.

The simulation results show the influence of the array element position deviation. Significant deterioration of radiation performance occurs for two typical antenna arrays. For a rectangular array, the antenna gain is reduced by 0.013 dB, the maximum sidelobe level is raised by 1.25 dB, and the main lobe beam pointing is shifted by a maximum of 2°. For the circular array, the maximum sidelobe level is raised by 4.84 dB, and the pointing error is extremely sensitive to the position error. The millimeter level position error can cause the antenna beam to point up to several degrees, which cannot meet the electrical performance requirements. To reduce the influence of position error on the electrical performance of the antenna, we should strictly control the position error of the array element during the machining process.

The model can calculate the electrical performance of the array helical antenna considering the machining position deviation, which provides theoretical guidance for antenna structure design.

## Figures and Tables

**Figure 1 sensors-23-04827-f001:**
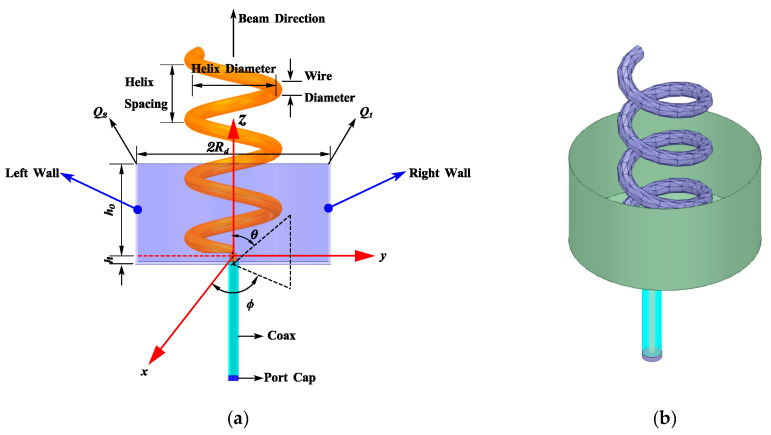
Helical antenna radiating cup and its coordinate system. (**a**) The geometric size of the radiation cup; (**b**) A three-dimensional model of the radiation cup.

**Figure 2 sensors-23-04827-f002:**
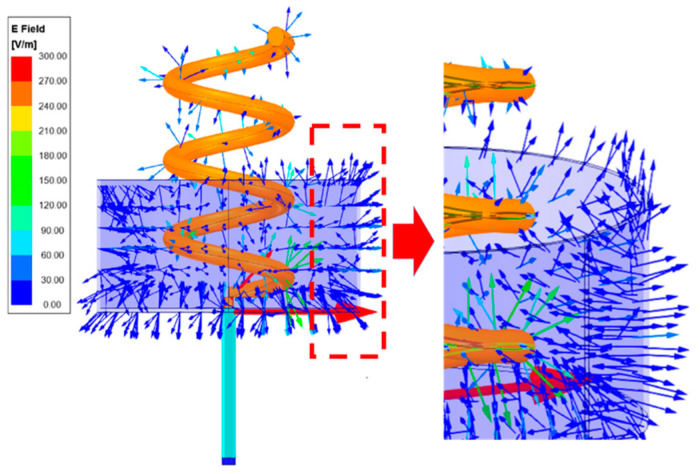
The vector distribution of the electric field at the edge of the diameter of the radiation cup.

**Figure 3 sensors-23-04827-f003:**
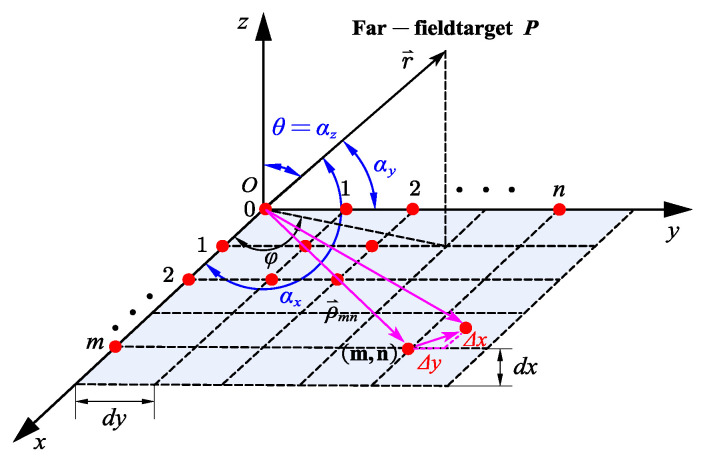
Schematic diagram of the array element arrangement of the rectangular.

**Figure 4 sensors-23-04827-f004:**
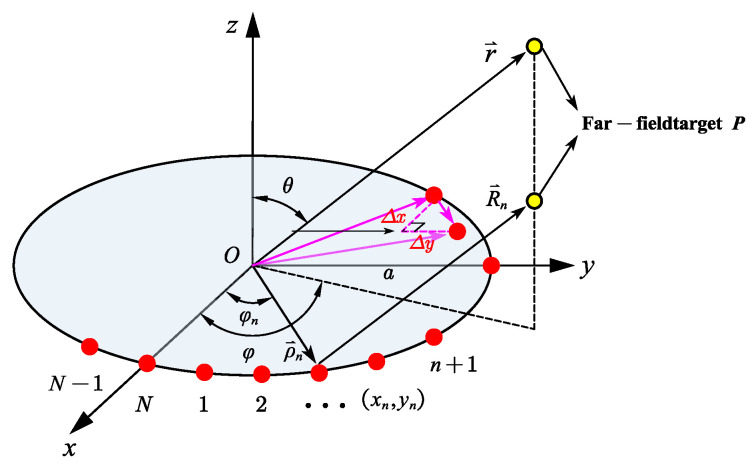
Circular array and its coordinate system.

**Figure 5 sensors-23-04827-f005:**
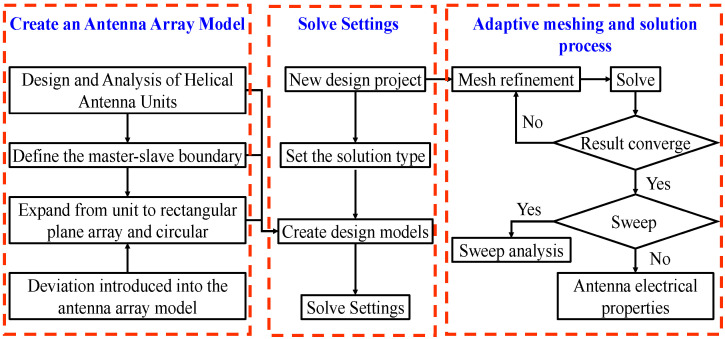
The calculation process of the electrical performance of the helical antenna array considers position deviation.

**Figure 6 sensors-23-04827-f006:**
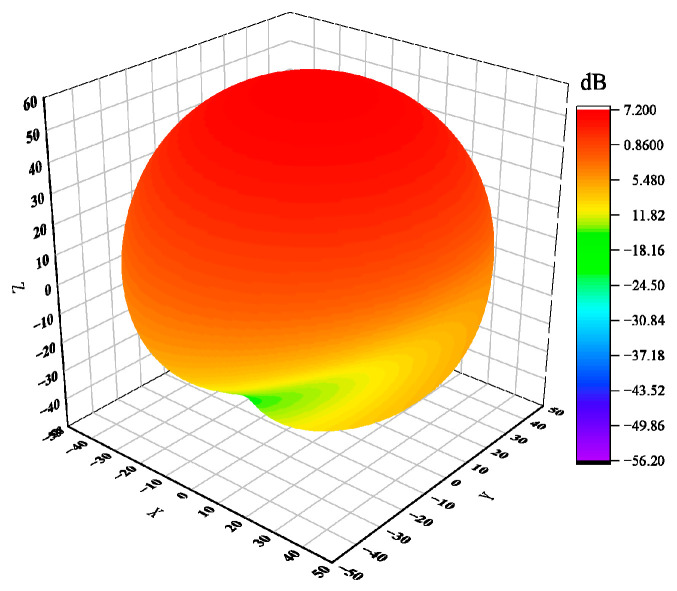
Three-dimensional orientation of an axial mode.

**Figure 7 sensors-23-04827-f007:**
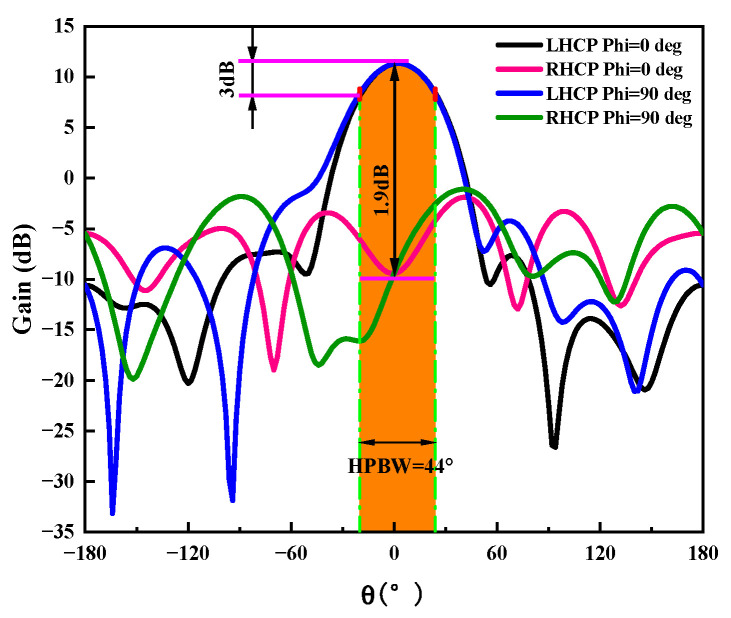
Radiation pattern at the center frequency.

**Figure 8 sensors-23-04827-f008:**
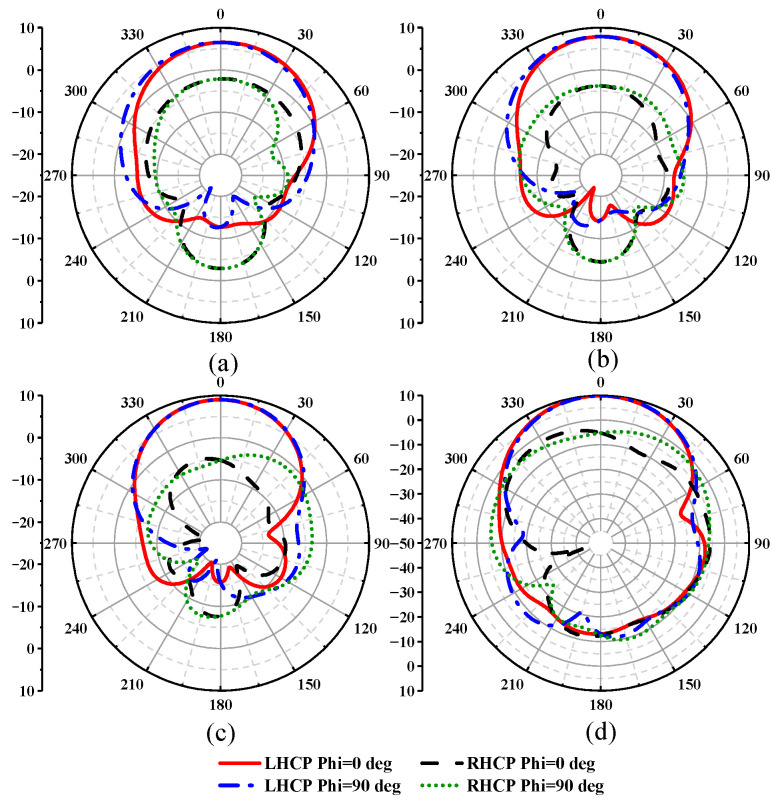
The pattern of the single winding axial helical antenna with a back cavity: (**a**) 1.2 GHz; (**b**) 1.4 GHz; (**c**) 1.6 GHz; (**d**) 1.8 GHz.

**Figure 9 sensors-23-04827-f009:**
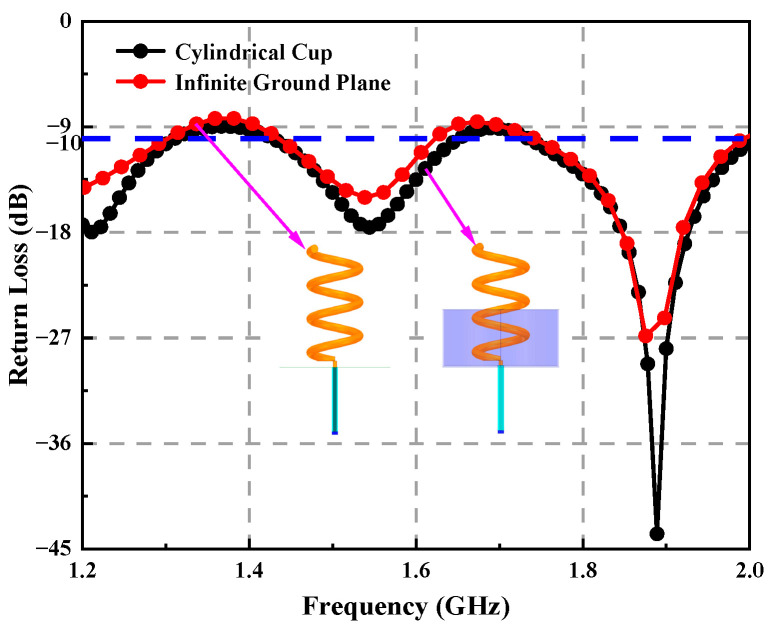
Return loss at different ground surfaces.

**Figure 10 sensors-23-04827-f010:**
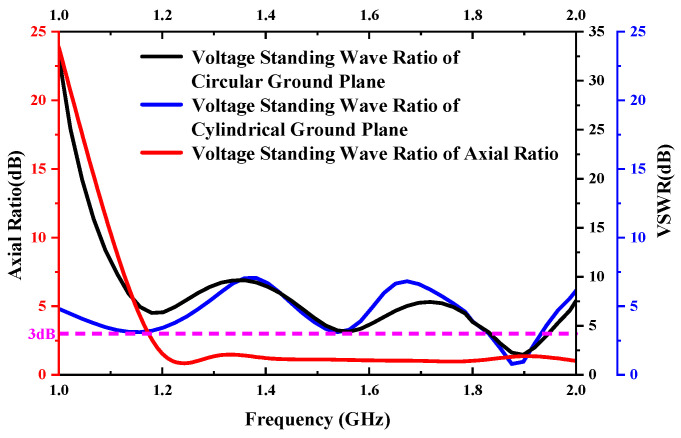
Difference on the surface of the earth axis ratio and standing wave ratio.

**Figure 11 sensors-23-04827-f011:**
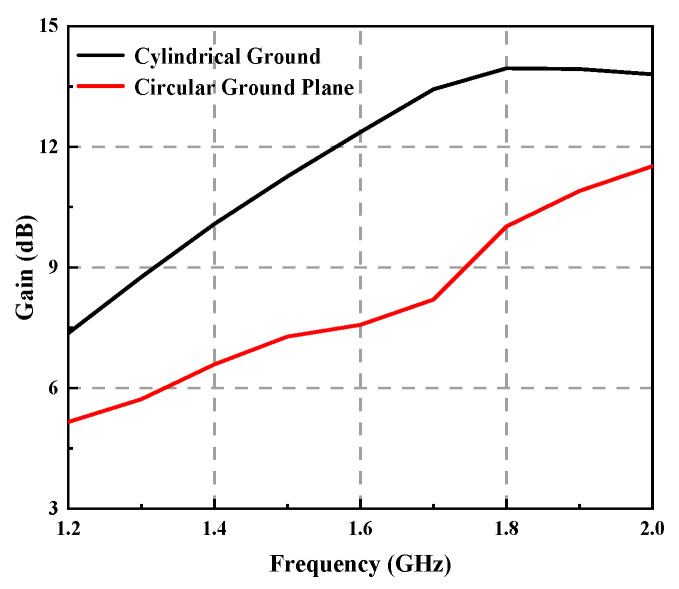
Gains of helical antennae with different ground planes.

**Figure 12 sensors-23-04827-f012:**
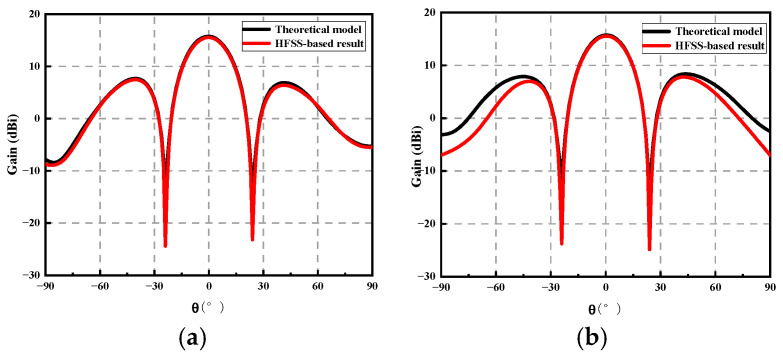
HFSS-based and this paper-based model results for comparison: (**a**) *ϕ* = 0°; (**b**) *ϕ* = 90°.

**Figure 13 sensors-23-04827-f013:**
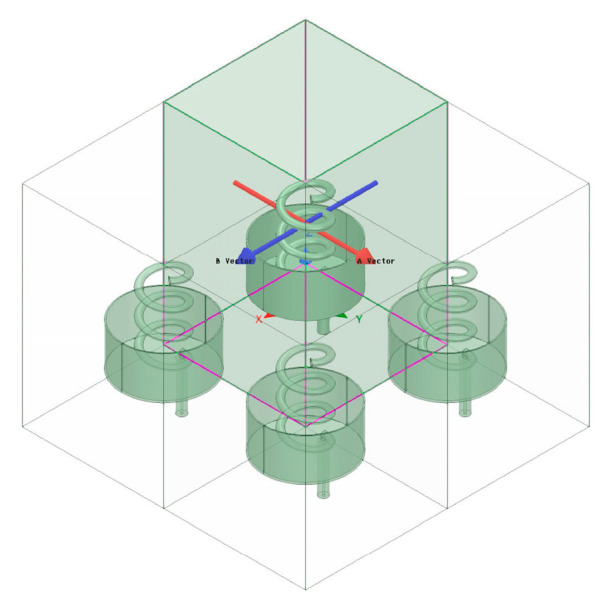
Schematic diagram of the rectangular planar array antenna.

**Figure 14 sensors-23-04827-f014:**
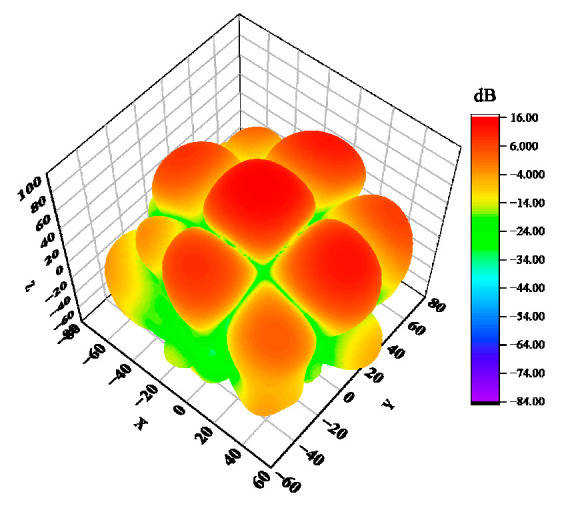
Three-dimensional orientation of rectangular array antenna.

**Figure 15 sensors-23-04827-f015:**
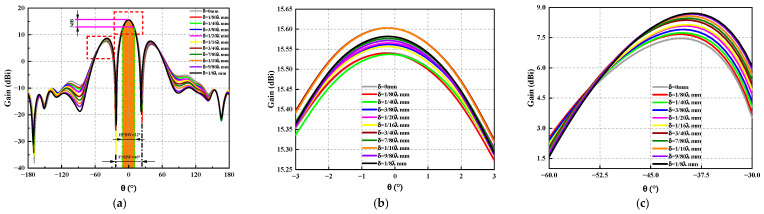
The radiation pattern of *ϕ* = 0° rectangular planar array antenna under different positional deviations. (**a**) The radiation pattern of an antenna. (**b**) The curve of the maximum gain of the antenna. (**c**) Change curve of antenna sidelobe level.

**Figure 16 sensors-23-04827-f016:**
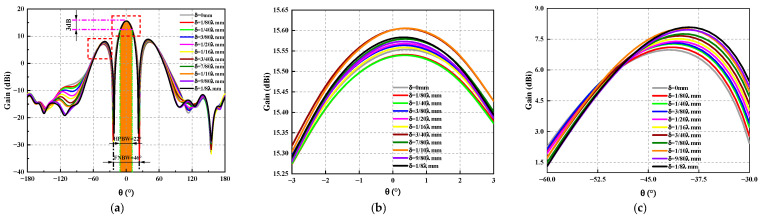
The radiation pattern of *ϕ* = 90° rectangular planar array antenna under different positional deviations. (**a**) The radiation pattern of an antenna. (**b**) The curve of the maximum gain of the antenna. (**c**) Change curve of antenna sidelobe level.

**Figure 17 sensors-23-04827-f017:**
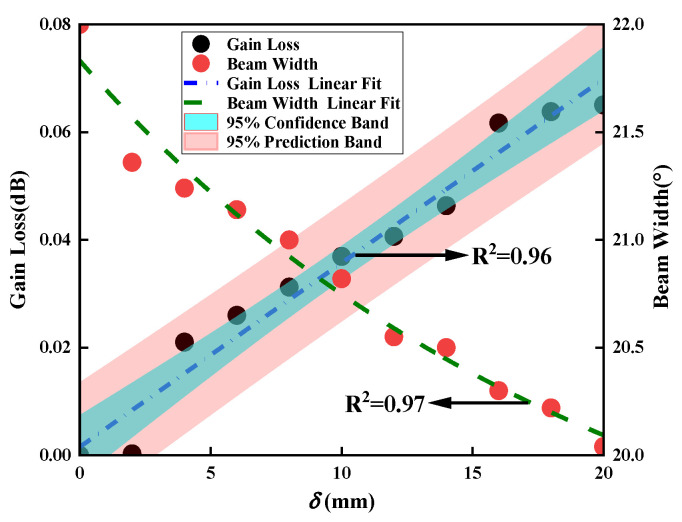
Antenna gain loss and beam width variation curves under different position deviations.

**Figure 18 sensors-23-04827-f018:**
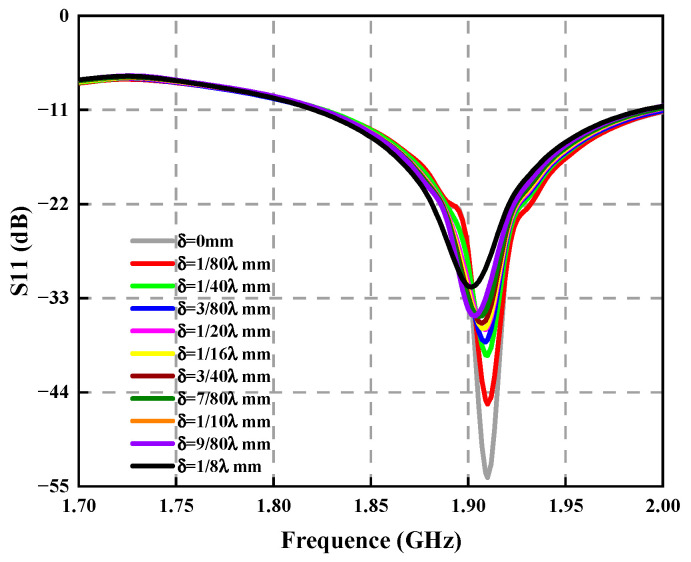
Variation curve of antenna return loss under different structural deviations.

**Figure 19 sensors-23-04827-f019:**
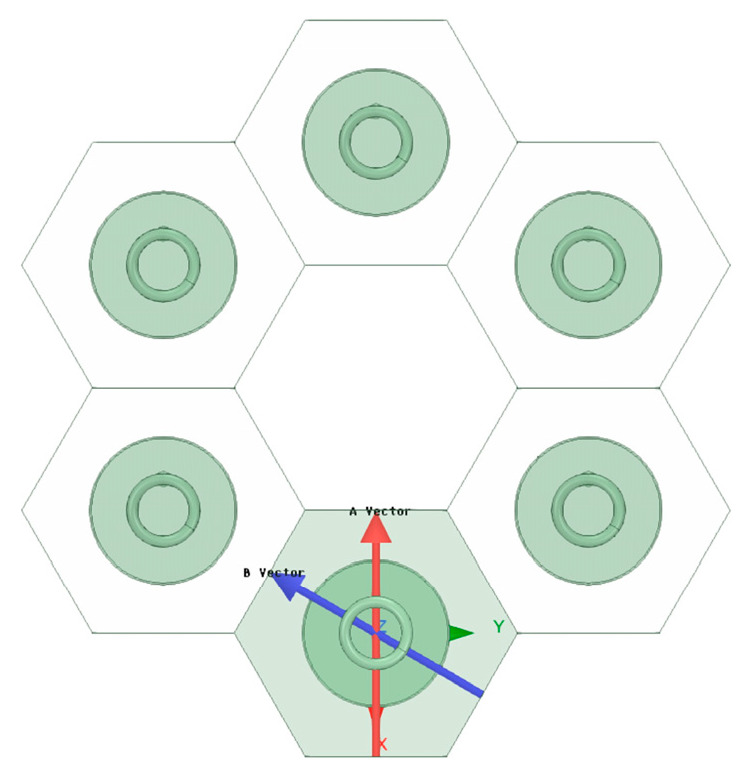
Schematic diagram of the circular ring array antenna array antenna.

**Figure 20 sensors-23-04827-f020:**
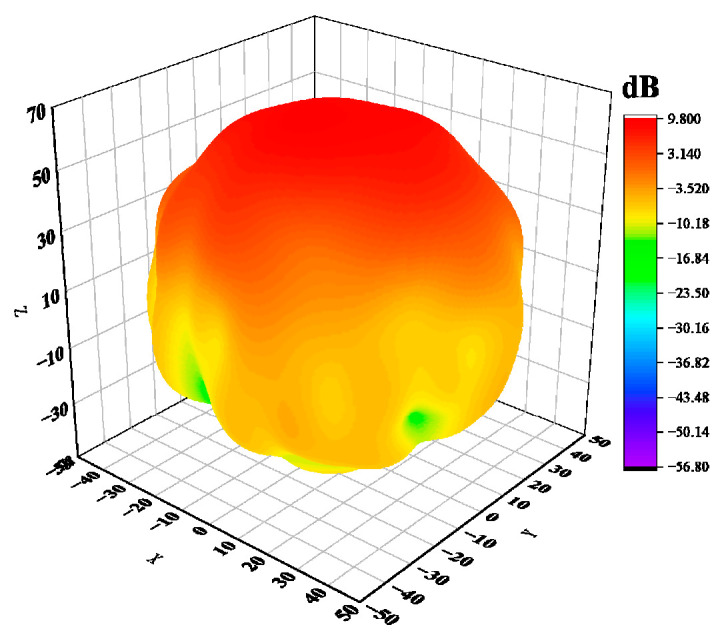
Three-dimensional stereogram of the circular ring array antenna.

**Figure 21 sensors-23-04827-f021:**
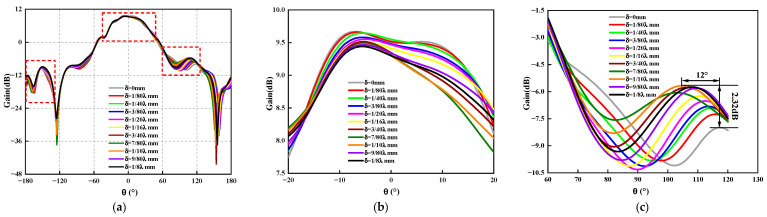
The radiation pattern *ϕ* = 0° of the circular ring array antenna under different position deviations. (**a**) The radiation pattern of an antenna. (**b**) The curve of the maximum gain of the antenna. (**c**) Change curve of antenna sidelobe level.

**Figure 22 sensors-23-04827-f022:**
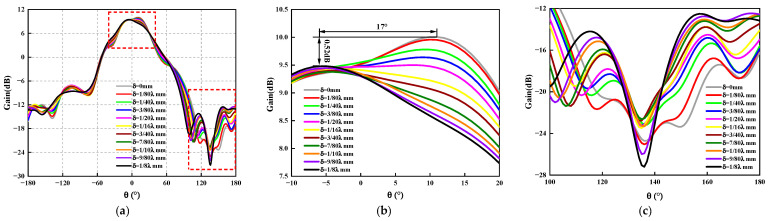
The radiation pattern *ϕ* = 90° of the circular ring array antenna under different position deviations. (**a**) The radiation pattern of an antenna. (**b**) The curve of the maximum gain of the antenna. (**c**) Change curve of antenna sidelobe level.

**Figure 23 sensors-23-04827-f023:**
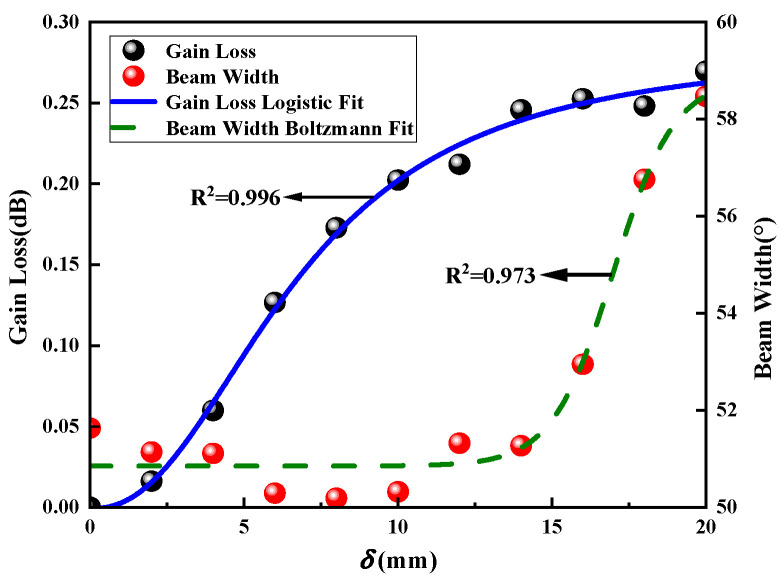
Antenna gain loss and beam width variation curves under different position deviations.

**Figure 24 sensors-23-04827-f024:**
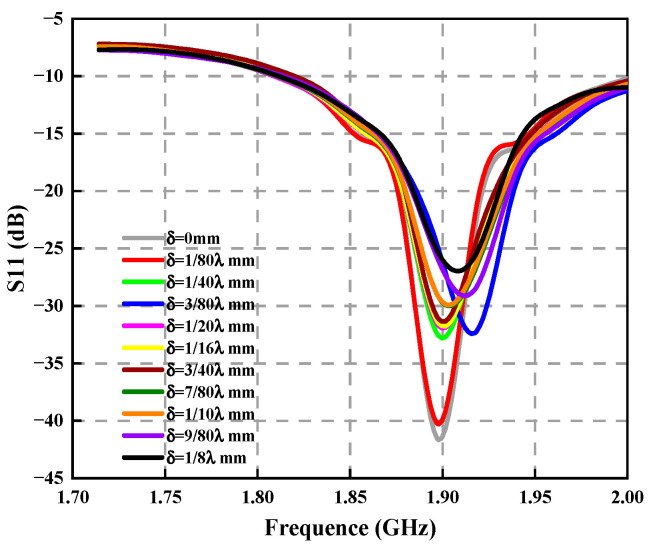
Variation curve of antenna return loss under different structural deviations.

**Table 1 sensors-23-04827-t001:** Structural parameters of axial-mode helical antenna.

Model Parameter	Symbol	$\mathrm{Value}\;(\lambda_{0}\;=\;1.89\;\mathrm{GHz})$
Diameter	*D*	63.61 mm
Circumference	*C*	πD
Spacing (center to center) between any two adjacent turns	*S*	44.32 mm
One turn of the screw length	*L*	$\sqrt{C^{2}+S^{2}}$
Diameter of the ground plane	$2R_{d}$	$0.75\lambda_{0}$
Diameter of the helix conductor	d	11.28 mm
Number of turns	*N*	3.341

## Data Availability

Not applicable.

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
