# Peer review of "Influence of Antenna Element Position Deviation on Radiation Performance of Helical Antenna Array"

_sensors, 2023, doi:10.3390/s23104827_

Round 1
Reviewer 1 Report
[1] Figure 1: Please draw a panoramic version of radiating cup instead of a rectangle.
[2] Line 132: The radiating cup is azimuth independent, and its wall surrounds the helix, please define the term “left/right sidewall”.
[3] Line 148: Please elaborate “the diffraction field generated by the reflection from the bottom”.
[4] Line 181: Please elaborate why put a “dot” above “I”.
[5] Lines 218-219: Please rephrase the following sentences: “The position deviation of the unit changes the time when the target signal arrives at the receiving unit, and the beam pointing of the antenna deviates from the ideal position.”
[6] Fig.5: Please elaborate how the diffraction analysis in Section 2 is fit in this flow-chart.
[7] Table 1: Suggest to add a column of “definition’’ or “parameter name”.
[8] Figure 10: Please clarify between “axial ratio” and “standing wave ratio”.
Grammar and writing style can be further improved.
Professional editing service is recommended.
Reviewer 2 Report
1. It appears many hyphenations in the title and text were misused?
2. The impact of array element position error on pattern's sidelobe level and mainlobe pointing are well-known. Please take feeding power constraint and manifold uncertainty constraint into account for enhancing the synthesis robustness against array element position error.
3. In Figs. 10, 11, 18, 21, 22, 24, the circle marker is redundant.
none.
Reviewer 3 Report
1, There is a lack of innovation with using the helical antenna instead of general antenna element, which lead to a narrow applicable area.
2, In this article, the array has 2 × 2 units, the number of element is too less to be persuasive
3, The deviation distance 0 to 1/8 wavelength has no actual manufacturing accuracy data support.
4, The mutual coupling between the different units is an important impact factors, which can not be easily excluded, especially for the helical antenna element.
5, It is not too comprehensive to uses the same excitation coefficient, which can not cover the the actual situation
The Writing English has no obvious grammatical errors ,and can express the author's meaning accurately
Reviewer 4 Report
The authros have proposed Influence of Antenna Element Position De-viation on Radiation Performance of Helical Antenna Array. A detailed discussion is presented. It shall be good to see one prototype being experimentally verified. further, how would the proposed investigation effect MIMO results on Helical/spiral shaped antenna strcutures? Can the authors comment with following references.
R. Johnson and R. Cotton, "A backfire helical feed," in IEEE Transactions on Antennas and Propagation, vol. 32, no. 10, pp. 1126-1128, October 1984, doi: 10.1109/TAP.1984.1143217. X. Dongyu, Z. Hou, Z. Qianyue and W. Chong, "Three Coaxial-feed Axial Mode Spherical Helical Antennas," The 2006 4th Asia-Pacific Conference on Environmental Electromagnetics, Dalian, China, 2006, pp. 468-471, doi: 10.1109/CEEM.2006.257997. L. Huang, J. Xiong and Y. Yu, "An electrically small normal-mode helical antenna with capacitive coupling feed," The 8th European Conference on Antennas and Propagation (EuCAP 2014), The Hague, Netherlands, 2014, pp. 2915-2917, doi: 10.1109/EuCAP.2014.6902436. A. N. Jaafar et al., "Analysis of Helical Antenna for Wireless Application at 2.4 GHz," 2019 IEEE Asia-Pacific Conference on Applied Electromagnetics (APACE), Melacca, Malaysia, 2019, pp. 1-5, doi: 10.1109/APACE47377.2019.9020810. T. M. D. Tran and M. Piette, "Bow-Spiral Antenna," 2019 IEEE International Symposium on Antennas and Propagation and USNC-URSI Radio Science Meeting, Atlanta, GA, USA, 2019, pp. 211-212, doi: 10.1109/APUSNCURSINRSM.2019.8888507.
Minor editing needed.
Round 2
Reviewer 1 Report
Most comments have been addressed.
Grammar and writing style can be further improved.
Author Response
Dear reviewer:
Thank you for taking your precious time to review our articles and put forward valuable comments and suggestions.We greatly appreciate your hard work and expertise. We modified some sentences and syntax, please see the attachment.
Best regards,
Lingqi Zeng

Reviewer 2 Report
The response is too simple. It is difficult for me to figure out how the manuscript was revised.
none.
Reviewer 3 Report
I have no further comments.
Author Response

(The authors gave the same response as above.)
